# Does Loading Ammonium to Sorbents Affect Plant Availability in Soil?

Bente Foereid * and Julia Szocs

Norwegian Institute of Bioeconomy Research Pb 115, NO-1431 Ås, Norway; juliaszocs@yahoo.com
* Correspondence: bente.foreid@nibio.no; Tel.: +47-401-05-960

**Abstract:** Sorption to cheap sorbents can be used to concentrate nutrients from liquid waste streams and make them into fertilisers. In this study we assess how plant available is ammonium nitrogen (N) sorbed to three sorbents, and if the potential for greenhouse gas (GHG) emissions after a non-growing season is affected by sorption. Ammonium-N labelled with N15 was sorbed to biochar, bentonite and zeolite. Treatments where N was sorbed and where N and sorbents were applied separately were tested in a pot experiment with wheat, and soil samples were then frozen and dried to simulate non-growing seasons. After thawing and re-wetting, GHG emissions from the soil were assessed. There was no difference between sorption treatments in biomass or N uptake or fertiliser N left in the soil, and little difference between sorption treatments in gas emissions after the non-growing seasons was seen. We conclude that ammonium applied sorbed to these sorbents is as plant available as ammonium applied the conventional way. GHG emissions at the beginning of the next season are also not affected by ammonium applied sorbed.

**Keywords:** ammonium sorption; bentonite; zeolite; biochar; greenhouse gases; plant availability





## 1. Introduction

To develop a circular economy, reduce greenhouse gas emissions, energy use, and resource mining, nutrients in waste streams should be recovered and brought back to agriculture to replace mineral fertilizers as much as possible [1]. Many waste streams contain diluted nutrients (e.g., sewage, digestate from biogas production, animal manure). Some water can be removed by dewatering, but most of the soluble and plant available nutrients are found in the liquid phase.

One way to concentrate nutrients in wastewater and waste with high water content would be to sorb it to cheap and abundant sorbents before dewatering [2–4]. Cation exchange is the most common, particularly removal of ammonium [3–6]. Examples of widely used sorbents are activated carbon, other charred materials ("biochar") [7–9], some minerals such as zeolite [10] and clay like minerals (e.g., bentonite, vermiculite) [11]. Biochar can be produced from locally available organic waste and also give energy in the production [12]. Zeolite [13] and bentonite [14] are found in sufficient quantities in many areas of the world.

The sorbent loaded with nutrients can then be applied to agricultural soils as fertilizer [15,16]. The application of sorbents may also improve soil properties in the longer term. However, if the application of sorbents pre-loaded with nutrients is to be effective as a fertilisation strategy, the sorbed nutrients need to be plant available within one growing season. This has not been much studied. Sorption of ammonium-nitrogen (ammonium-N) to zeolite seems to make a slow-release fertiliser [17–19]. Our previous studies also suggest sorption of ammonium-N sorbed to zeolite makes it less available to wheat [20]. Ammonia gas sorbed to biochar was plant available [21], but Kocatürk-Schumacher et al. [16] found that zeolite was a better sorbent than biochar. We have not found any information on any other sorbents.

Greenhouse gases (GHG's) are emitted from agricultural soil. The most important in non-flooded systems is $N_2O$. There are a number of mechanisms $N_2O$ can be produced by [22], and it is therefore difficult to predict emissions. Particularly, emissions from organic residues may be high and unpredictable [23–26], as can pulses after a non-growing season, particularly freezing–tawing cycles [27–31]. Low utilisation of N by the previous crop can give larger losses during freeze-thaw [32]. Mixing sorbents into ammonium-rich residues before application to soil have also been reported to reduce gaseous losses of N [33–35]. It is uncertain if this effect can also be seen the next season.

Here we test three sorbents for ammonium sorption from liquid and assess how plant available the sorbed ammonium is. We also measure GHG emissions after a simulated winter and dry season. Our hypothesis was that (1) N applied sorbed would be less plant available the first season than that applied not sorbed and (2) sorbed application would leave more N available for $N_2O$ emission the next season.

## 2. Materials and Methods

### 2.1. Sorbents and Soil Used

The zeolite was the same as used by Foereid et al. [20]. It was from a mine in Bodrogkeresztúr, Hungary, marketed by Colas Hungária Ltd. It was of Mahdi type, specific surface area (BET method) 33.14 $m^2 g^{-1}$, composed of Montmorillonite (30–45%), Varco (30–40%), Kaolinite (10–15%), Feldspar (3–7%) and Calcite (1–2%). The bentonite (Montmorillonite content ~50%) was sampled on 2016 from a mine located in Pétervására, Hungary (48.023° N 20.099° E). Its chemical composition was determined by fluorescence X-ray spectrometry (XRF) from scanning electron microscopy (SEM) in weight % (O 61%, Si 26%, Al 7.2%, Fe 1.8%, K 0.9%, Ca 0.9%, Mg 0.8%, Na 0.7%, Ti 0.2%). The biochar was the same as used by Foereid et al. [32]. It was produced by PYREG GmbH in Germany by pyrolyzing *Miscanthus giganteus* at 600C (C: 79.6%, H: O: 8.0%, 0.47%, N: 0.31%, pH 7.86). It has previously been found that this biochar has no N fertilizer effect (Foereid, unpublished).

The soil was the same as that used by Foereid et al. [32] and measured soil parameters can be found there. It was an agricultural soil from Øsager experimental field in Østfold in eastern Norway (59.31936° N, 11.04221° E). It was a stagnosol, clay loam (clay 42%, silt 44%, sand 13%) [36]. It was collected in November 2017 and stored in an unheated cellar until start of the experiment in February 2018.

### 2.2. Mode of Sorption

Initial tests were performed with five sorbents, bentonite, zeolite, alginate, and two types of biochar (from wood and miscanthus). Sorption capacity was determined by shaking for three days in a 1 gN/L solution of ammonium sulphate, and concentration was measured colorimetrically (Spectroquant Pharo 100) in the remaining solution. The three sorbents with the highest sorption capacity were selected, zeolite, bentonite, and miscanthus biochar. The amount of each sorbent used in each pot treated with that sorbent was the amount needed to sorb 0.2 g N which was the amount of N needed per pot for optimal N fertilisation. The parameters for treatment are given in Table 1. Basically, a double strength solution (2 gN/L) was used for biochar, otherwise shaking for three days was carried out. Biochar was separated by filtering, the two others by centrifugation (Table 1). When the procedure had been established, the loaded sorbents for the growth experiment were prepared with [15]N labelled ammonium sulphate labelled at 5% strength in all treatments receiving N.

### 2.3. Growth Experiment

The treatments with sorbents are shown in Table 1. In addition, there were two treatments without sorbents one with no N fertilization (0N) and one with full N fertilization using ammonium sulphate (1N). The same amount of total N, 0.2 g per pot was given to all treatments receiving N. Phosphorus and potassium (0.26 g $KH_2PO_4$ and 0.43 g $K_2SO_4$) was added to all treatments. Soil and additions were mixed thoroughly in each pot, and

water content adjusted to half field capacity. There were three replicates of each treatment. Wheat (variety "Bjarne") was used as a test plant. The pot size was 2 L and 15 seeds were sown in each pot, thinned to 10 shortly after germination. The temperature in the greenhouse was kept at or above 20/12 °C day/night and 16 h daylength. Plants were watered to keep water content between half and full field capacity, and pots were moved around in a random manner each time they were watered. Plants were harvested just after ear emergence, after 50 days of growth. Aboveground plant material was cut just above the soil with scissors. Plant samples were dried (70 °C), ground into a fine powder and weighed in for analyses of total N and C. Total N and C were measured on a CHN analyser (Elementar Vario EL with TCD detector) [37]. Soil and plant samples were sent to UCDavis (https://stableisotopefacility.ucdavis.edu/ accessed 15 May 2022) for analysis of isotope ratio. Unlabelled samples were included for comparison. pH in the soil was also measured after the growth period.

**Table 1.** Sorbents and sorption treatments used. Treatment without sorption means sorbent and N both applied separately to the soil.

|  | **Zeolite** | **Bentonite** | **Biochar** |
|---|---|---|---|
| Origien | Hungary | Hungary | Produced in Germany |
| Pre-treatment | Dried 70C overnight | Dried 70C overnight | |
| Amount of sorbent | 34.30 g | 34.46 g | 21.71 g |
| Sorption treatment | ZE-SN | BE-SN | BI-SN |
| Treatment without sorption | ZE+N | BE+N | BI+N |
| Sorption condition | 1:10 sorbent: liquid 3 days, 1 g N/L | 1:10 sorbent: liquid 3 days, 1 g N/L | 1:10 sorbent: liquid 3 days, 2 g N/L |
| De-watering | Centrifuge 4000 rpm, 15 min | Centrifuge 4000 rpm, 15 min | Filtering in coffee filter |

*2.4. Incubation and GHG Measurements after a Non-Growing Season*

To simulate a non-growing season (winter or dry season), one sample of 20 mL of soil was taken out of each pot and air-dried and another sample was frozen ($-20$ °C). After approximately three months they were taken out of the freezer or re-wetted close to field capacity. The 0N samples were excluded from this incubation. Samples were kept in 100 mL bottles that were put in an incubator at 20 °C and 90% relative humidity. For measurement of the gas produced, the bottles were closed for $\pm1$ h. Then 12 mL of gas was extracted through a septum in the closed lid with a syringe and injected into an evacuated vial. A zero sample was taken from the top of the open bottles before closing. Bottles were then opened and kept open during the experiment. The experiment ran for 4 days.

The gas samples were analysed by gas chromatography mass spectrometry (GC-MS) to determine concentrations of nitrous oxide ($N_2O$), methane ($CH_4$), and carbon dioxide ($CO_2$). The analysis was performed using an Agilent Technologies 7820A GC System gas chromatograph, coupled to a mass detector Agilent Technologies 5875 Series MSD and a Gilson 222 XL auto sampler. The sample was injected by a 5 mL sample loop, through a 0.5 m $\times$ 0.32 mm deactivated precolumn, into a 25 m $\times$ 0.32 mm CP-PoraPLOT Q-HT column (Chrompack), kept at 40 °C. Helium was used as carrier gas at 1.0 mL min$^{-1}$.

On day one and day three, a separate sample was extracted after closing the bottles the same way a second time. The vials were sent to UCDavis stable isotope lab and analysed for 15N in $N_2O$. Unlabelled samples were included for comparison.

*2.5. Calculations and Statistics*

The fraction of N originating from the applied N in soil, plant and $N_2O$ was calculated assuming no fraction.

Statistical analysis was performed with Minitab v18, ANOVA to determine if there was any effect of factor (e.g., fertiliser addition mode). When the effect of fertiliser addition mode was assessed, the treatment without N addition was excluded from the statistical analysis.

Differences between individual treatments were analysed with t-test when appropriate. Data from the incubation were analysed separately for each measurement point.

## 3. Results

Biomass drymatter with fertiliser was on average 42% higher than without (Figure 1). On average 59% of the fertiliser taken up by the fertilised plants came from the fertiliser (Figure 2). N uptake in the unfertilised control was only 53% of average uptake from soil in the fertilised treatments (Figure 2). Only the 0N treatment was significantly different from the rest, otherwise there was no significant difference in either biomass (Figure 1) or N uptake (Figure 2) from either fertilizer or soil between fertiliser addition modes and sorbents. There was a significant effect of fertiliser addition or sorbents on biomass. Uptake with N applied sorbed to biochar (BI+N) was significantly different from 1N. When no fertiliser addition was included in the ANOVA, there was a significant effect of treatment on uptake from soil. All the treatments where N was applied sorbed as well as ZEO-SN and 1N were significantly different from no N fertiliser. There were minor changes in soil pH during the growth experiment (data not shown) and there were differences between treatments. However, these differences were not large enough to cause changes in plant growth [38].

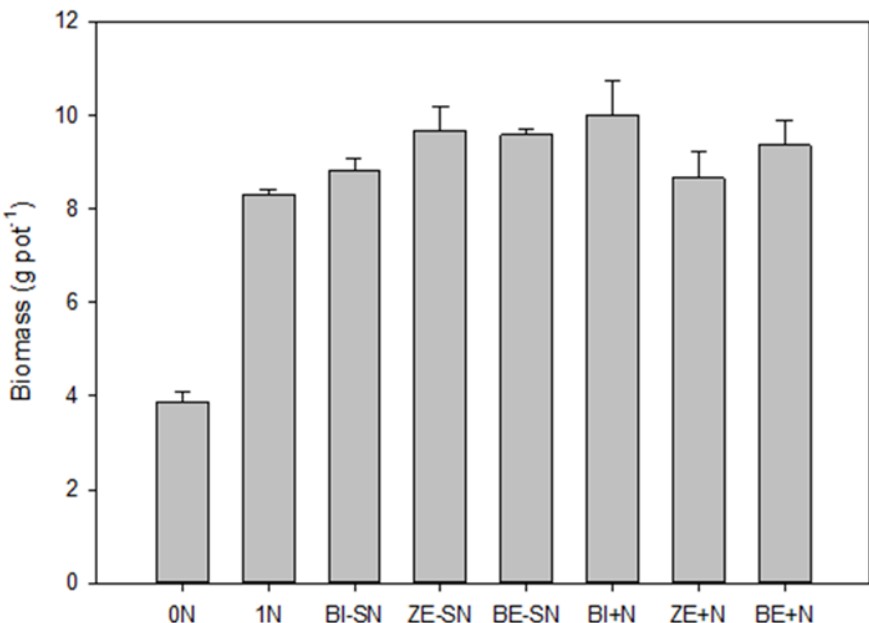

**Figure 1.** Plant biomass for each treatment at harvest. Error bars are standard error (n = 3). 0N = no fertilizer, 1N = full mineral N fertilizer, no sorbent, BI = biochar sorbent, ZE = zeolite sorbent, BE = bentonite sorbent. When sorbent and N fertilizer are both mixed into the soil separately, -SN is added. When N fertilizer is applied sorbed with the sorbent +N is added.

There was no significant difference in the fraction of N left in the soil after the experiment between the modes of N application and sorbents (Figure 3). There was also no methane emission (data not shown). In the second last measurement of $CO_2$ after drying, there was a weak significant difference from the rest. Otherwise, there were no significant differences between treatments in either $CO_2$ (Figure 4) or nitrous oxide (Figure 5) emissions, but there were large differences between emissions after a dry period and after freezing. Soil respiration ($CO_2$ emission) was of similar magnitude just after freezing and drying, but after freezing it soon fell to very low levels whilst after drying it stayed in the same range (Figure 4). $N_2O$ emissions were very low after freezing, whilst after drying they were similarly low for a few hours, and then rose (Figure 5). The 15N signature in $N_2O$ after a freezing or drying treatment is shown in Figure 6. There was a marginally significant effect of sorption treatment the first day in dried treatment, otherwise there was

no significant effect of sorption treatment. But a much larger fraction of N2O originated from the applied fertiliser after a dry period than after freezing.

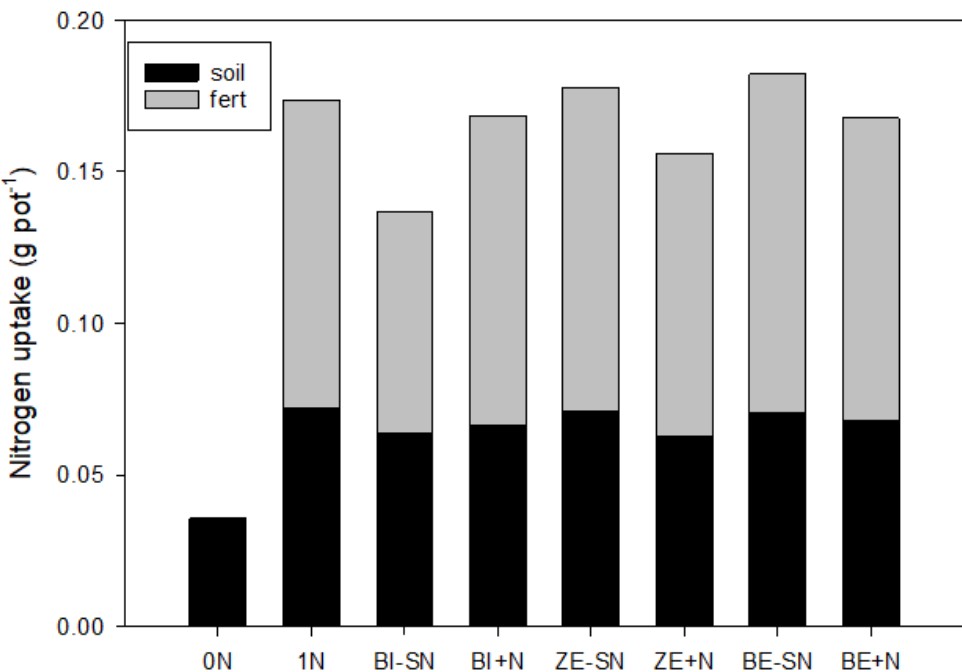

**Figure 2.** Plant N uptake from soil (black) and from applied fertilizer (grey) calculated from 15N signature. 0N = no fertilizer, 1N = full mineral N fertilizer, no sorbent, BI = biochar sorbent, ZE = zeolite sorbent, BE = bentonite sorbent. When sorbent and N fertilizer are both mixed into the soil separately, -SN is added. When N fertilizer is applied sorbed with the sorbent +N is added.

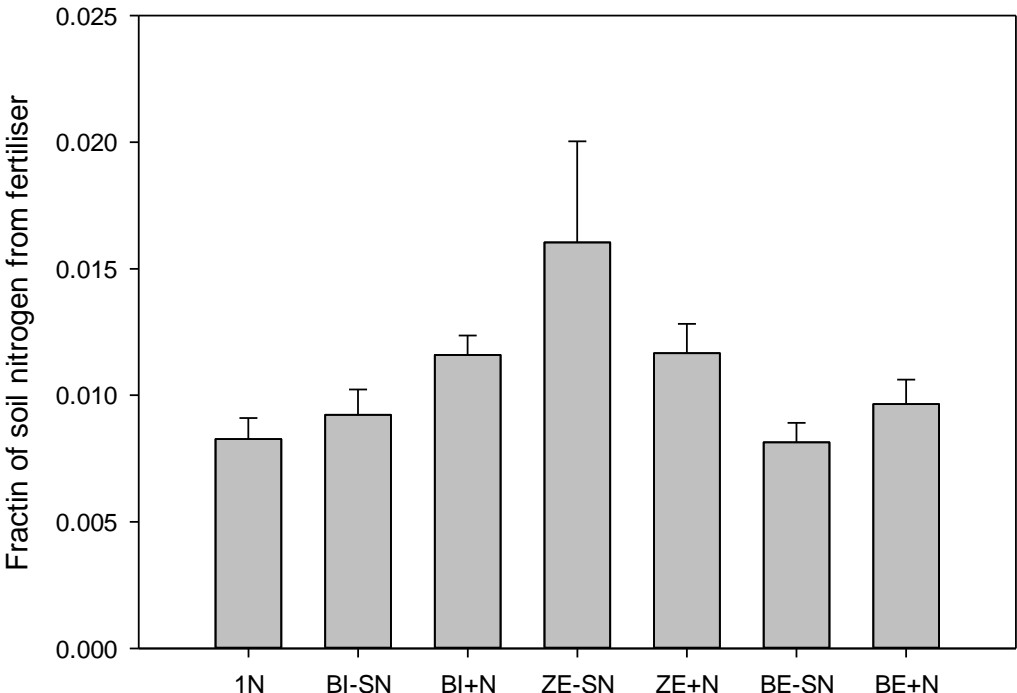

**Figure 3.** Fraction of soil N from applied fertiliser after plant growth, calculated from 15N signature. Error bars are standard error (n = 3). 1N = full mineral N fertilizer, no sorbent, BI = biochar sorbent, ZE = zeolite sorbent, BE = bentonite sorbent. When sorbent and N fertilizer are both mixed into the soil separately, -SN is added. When N fertilizer is applied sorbed with the sorbent +N is added.

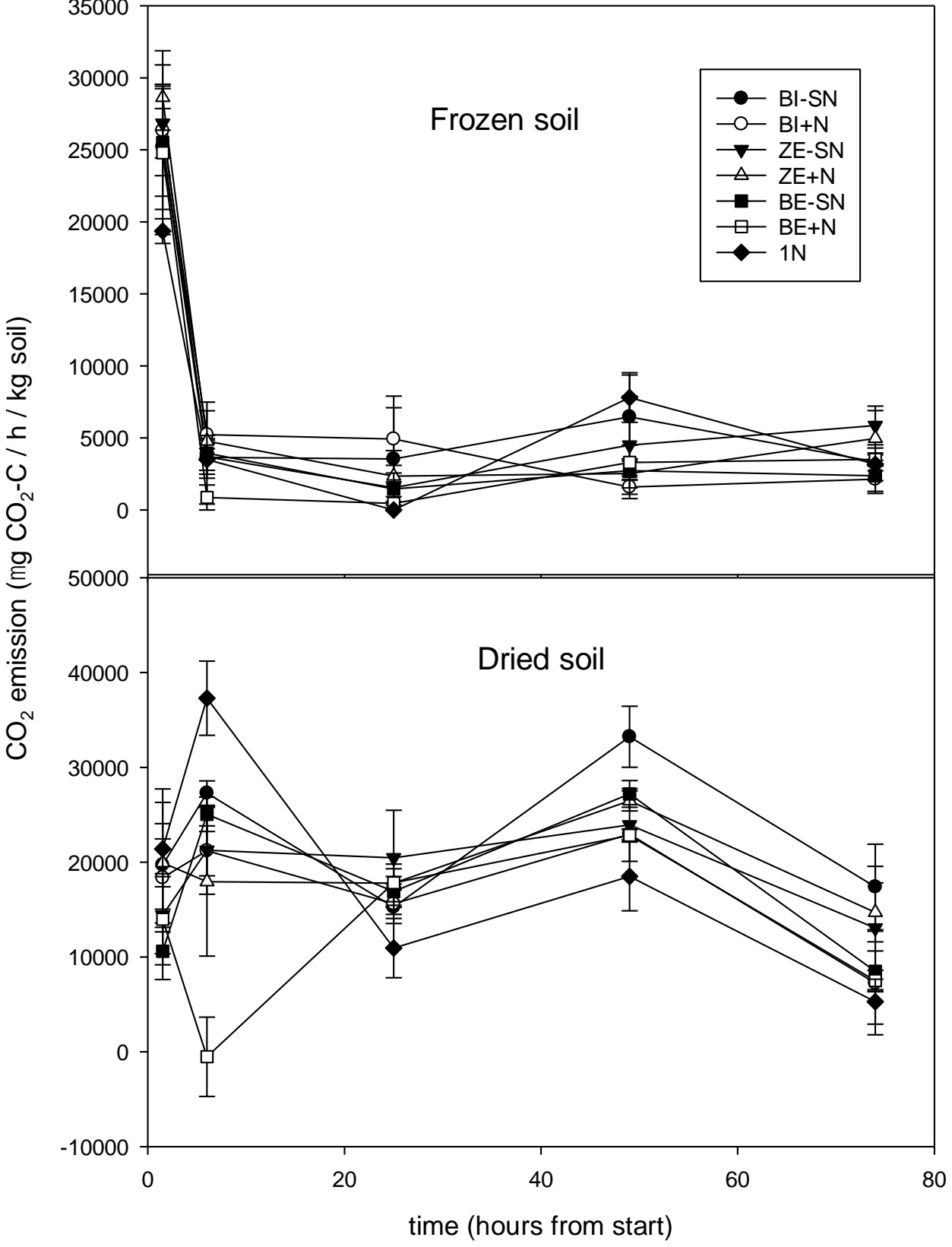

**Figure 4.** $CO_2$ emission from soil after thawing and re-wetting. Error bars are standard error (n = 3). 1N = full mineral N fertilizer, no sorbent, BI = biochar sorbent, ZE = zeolite sorbent, BE = bentonite sorbent. When sorbent and N fertilizer are both mixed into the soil separately, -SN is added. When N fertilizer is applied, sorbed with the sorbent +N is added.

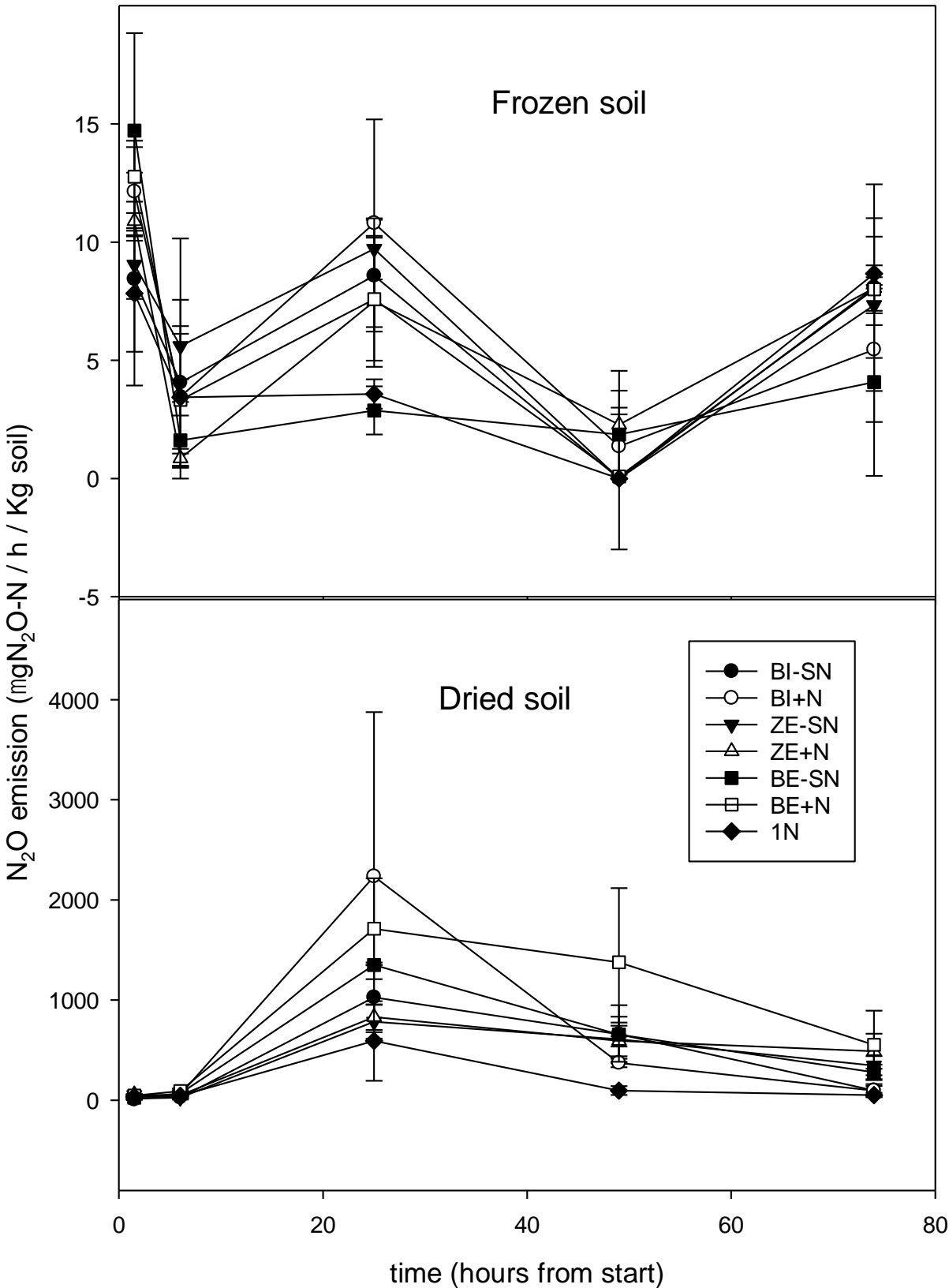

**Figure 5.** $N_2O$ emission from soil after thawing and re-wetting. Error bars are standard error (n = 3). 1N = full mineral N fertilizer, no sorbent, BI = biochar sorbent, ZE = zeolite sorbent, BE = bentonite sorbent. When sorbent and N fertilizer are both mixed into the soil separately, -SN is added. When N fertilizer is applied, sorbed with the sorbent +N is added.

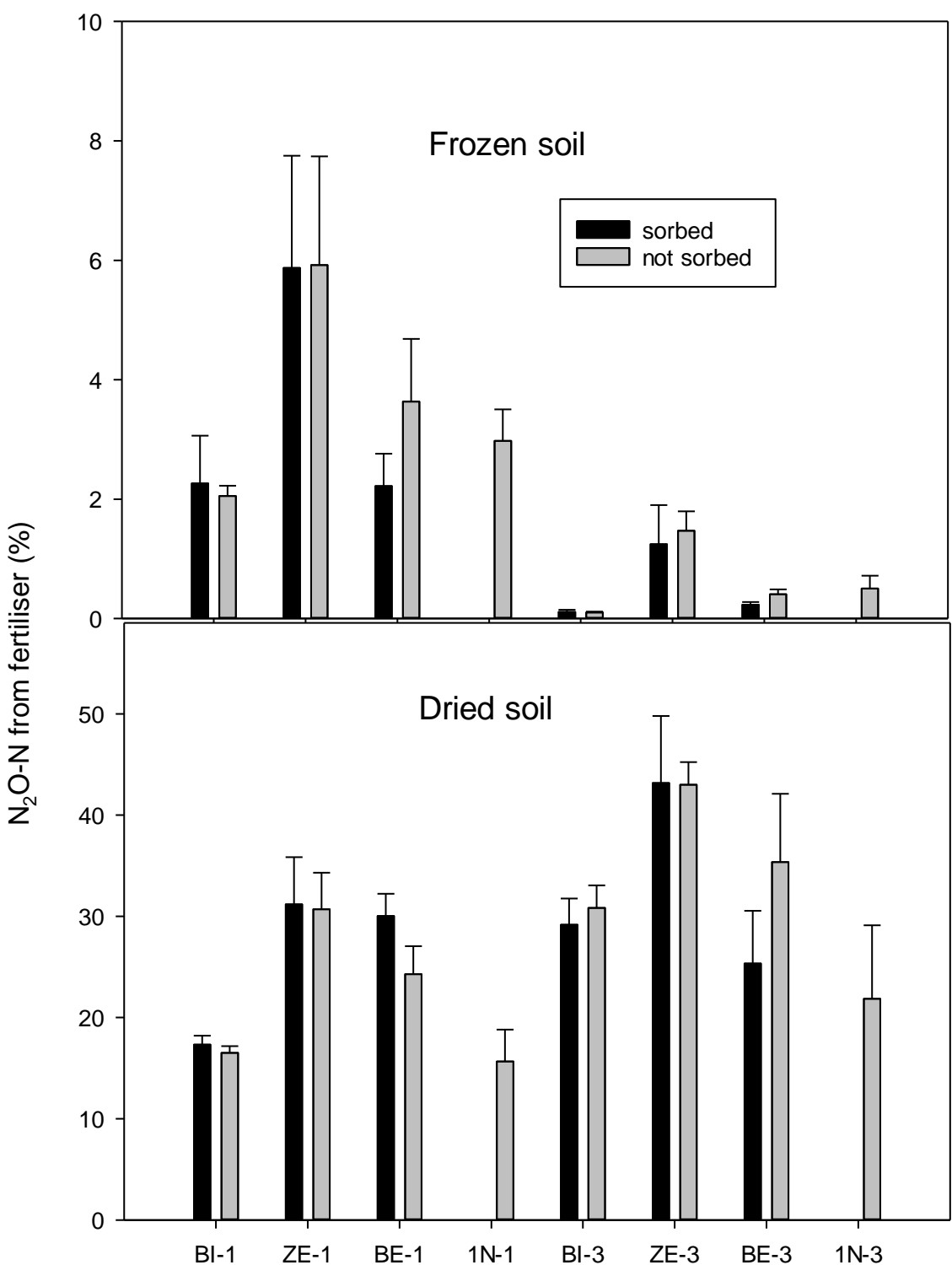

**Figure 6.** Percentage of N$_2$O -N originating from the applied fertiliser on day 1 (shown as -1) and day 3 (shown as -3) of the incubation. Error bars are standard error (n = 3). BI = biochar sorbent, ZE = zeolite sorbent, BE = bentonite sorbent. -1 means day 1, -3 means day 3.

## 4. Discussion

Applying N sorbed to any of these sorbents did not appear to affect plant uptake of N or N transformation in the soil to any significant degree. This indicates that ammonium applied sorbed is equally plant available to ammonium applied the conventional way. The type of sorbent does not matter for how plant available it is afterwards, but there were

differences between sorbents in how easily they were sorbed. Biochar appeared to have the least capacity. A previous study has also indicated that the expectations to biochar have been too high [39], but chemical treatments may improve sorption capacity [40]. It can therefore be recommended to use other sorbents such as zeolite or bentonite when they are available, but biochar has the advantage that it is available everywhere. The use of these sorbents can be recommended to take out nutrients from liquid waste streams and to mix with manure etc., before application [3,4,15,32,40].

The results here are in contrast to earlier results [32] where it was found that sorption to zeolite did reduce subsequent plant availability. In that study, two contrasting soils, both different from the one used here, were used, and it was found that soil type was an important factor. It is therefore possible that sorption could have an effect on other soil types. However, in that study, zeolite with sorbed ammonium was also dried before use and some ammonium could have been lost during drying, whilst in the present study the loaded sorbents were used wet, shortly after sorption. This could indicate that sorbed ammonium is not as stable as the literature indicates [41], or at least a fraction of the ammonium that can be taken out with the sorbent is easily lost.

The treatment without N application took up less N also from soil than fertilised plant. This has also been found previously [20,32,42]. It appears that severely N limited plants are not able to take up N as efficiently as well fertilised plants, possibly because they are not able to produce enough roots.

The study indicates little or no effect of N application mode or sorbents on N left in the soil after the growing season or GHG emissions after a non-growing season. This can be explained by little or no difference in N uptake the season before. Foereid et al. [32] found that low N utilisation the season before can induce much larger $N_2O$ emissions during freeze-thaw the next season. However, there was a large difference between the effects of a dry period and a frozen period. Although this soil was from a temperate region and the microbial community would have been adapted to frost every winter, freezing had a much stronger effect on subsequent respiration and $N_2O$ emission than drying, as has been found previously [28,32]. The high respiration rate just after thawing can be explained by microbial biomass killed and ruptured by freezing was being utilized by other microbes [43–45]. However, this was quickly used up, and then respiration rate fell, and did not recover over the few days when the study was conducted. After drying however, respiration rate recovered immediately upon re-wetting. We should remember that this was a very extreme winter with stable $-20\,^\circ$C and then very rapid thawing. In most climates there would be multiple freezing-thawing cycles each year. This may increase the yearly total $N_2O$ emission from residues [27]. More experiments simulating more realistic winter and spring conditions are recommended.

## 5. Conclusions and Perspectives

The study suggests that N applied sorbed is as available as N applied conventionally, and there is also little or no effect on GHG emissions the next season. Use of sorbents to fix ammonium from diluted waste streams can therefore be recommended. As there is also evidence in the literature that also other nutrients are sorbed and desorbed in a similar manner [46], this could be a good method to concentrate nutrients from liquid waste streams. Not all the nutrients can be recovered this way, but this method could be used before other standard treatments (e.g., precipitation etc.). Sobents can also be mixed with manure, digestate etc., before application to soil to reduce nutrient losses.

There is also no indication that the type of sorbent matters much for desorption in soil, and much of the extensive research carried out on biochar will probably be relevant also for some other sorbents. However, more experiments with different soil types are recommended. It is likely that the effect will be best in soils with little sorption capacity, and in such soils also a residual benefit of leaving the sorbent in the soil can be expected.

**Author Contributions:** Conceptualisation, B.F.; methodology, B.F.; investigation, J.S.; resources, B.F.; data curation, B.F.; writing—original draft preparation, B.F.; funding acquisition, B.F. All authors have read and agreed to the published version of the manuscript.

**Funding:** This research was funded by Norwegian Research Council project SIS—Sustainable recycling of organic waste resources in the future bioeconomy and Biogas 2020 Interreg Øresund-Kattegat-Skagerrak project.

**Institutional Review Board Statement:** Not applicable.

**Informed Consent Statement:** Not applicable.

**Data Availability Statement:** The data presented in this study are available on request from the corresponding author.

**Acknowledgments:** The authors wish to thank Monica Fongen and Jan Erik Jacobsen for analysis work.

**Conflicts of Interest:** The authors declare no conflict of interest.

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
