# Peer review of "Does Loading Ammonium to Sorbents Affect Plant Availability in Soil?"

_agriculture, doi:10.3390/agriculture12071057_

Round 1

Reviewer 1 Report

The manuscript entitled “Does loading ammonium to sorbents affect plant availability in soil?” is complete and well within the scope of this journal. This study investigated the use of affordable sorbents to concentrate nutrients from liquid waste streams and make fertilisers. This MS is well-written and interesting to the readers. However, this MS has some major issues that need to be addressed.

Comments

Why were wheat plants selected for the experiment?

The authors should have explored the treatment effects at various plant development stages.

In the figure legends, treatment abbreviations and treatment details should be included.

Discussion: Relevant and current references should be included to improve the interpretation of the findings.

Future perspective should be included to strengthen the conclusion.

Author Response

Why were wheat plants selected for the experiment?

There is no particular reason, but it is the most common crop to use for these types of experiments, and that makes comparison with other experiments easier. It is also one of the world’s most important crops.

The authors should have explored the treatment effects at various plant development stages.

Yes, maybe that would be nice, but it would be much more work. And it is too late now….

In the figure legends, treatment abbreviations and treatment details should be included.

This has been included, as also recommended by the other reviewer.

Discussion: Relevant and current references should be included to improve the interpretation of the findings.

Some more discussion including newer references has been added.

Future perspective should be included to strengthen the conclusion.

This has been included as far as possible.

Reviewer 2 Report

Please check the yellow marked points in the revised text.

Author Response

We wish to thank this reviewer for detailed help in spotting formatting errors.

Reviewer 3 Report

Dear Editor

I write in reference to the manuscript “Does loading ammonium to sorbents affect plant availability in  soil?  by Foered and Szocs. The manuscript refers to an experiment using three different sorbents to load ammonium and fertilize a specific soil where wheat was grown. The results show that direct application of ammonium or use of  sorbents produce basically the same result. It could be interesting because sorbents may be a cheap eco-friendly form to recuperate nitrogen from alternative sources.

Although the manuscript has several problems that need to be corrected .

Specific concerns.

Lines 63 and 70. Please provide the manufacturer, address and catalog number for both products.

Line 66 and 74. Please provide the coordinates of both sites.

Lines 82-83. Please describe the method or provide an appropriate reference.

Line 89. Please explain which are the characteristics of "Treatment without sorption" (table 1). Had ammonium these treatments? In what form was loaded?

Line 100. Was ammonium sulfate isotopically marked? Please explain. If it is the case, please provide the source of marked ammonium. 

Lines 111-112. “Total N and C was measured on CHN analyzer (Elementar Vario EL with TCD detector). Soil” Please describe the method or provide an appropriate reference.

Lines 112-113. “Soil and plant samples were sent to UCDa-vis for analysis of isotope ratio. Unlabelled samples were included for comparison”.  Please describe with detail the methods of analysis or provide an appropriate reference. Please explain the origin of the isotope differences.

Line 120. capacity. “The 0N samples were excluded from this incubation” Why?

Line 122-125 and 133-134. Were samples taken with the bottles closed or opened? It is very confusing. Please rephrase and provide an appropriate reference.

Line 140. Please indicate which PosHoc probe was used.

Line 142 “the treatment without N addition was excluded from the analysis”. Why?  Please explain which treatment, (Fig 2 only?).

Line 147 . Please explain (in M&M section) how was determined the origin of the nitrogen.

Lines 151-153. “When uptake from fertiliser was analysed separately, there was a significant effect of fertiliser addition and 152 sorbents.” . Does It means eliminating 0N from the statistical analysis? It would be incorrect. Please explain. 

Lines 151 156. Please rephrase, it is very confusing. 

All the figures, “N=3”. Please use "n" (lowercase).

Figure 1. Please use letters on the bar tops to indicate statistical differences and add the explanation in the caption including the statistical analysis employed.

Figure 2. Why aren't error bars included? Were there no replicates?

Line 167. “Figure 2” must say Figure 3. Please correct. Please use letters on the bar tops to indicate statistical differences, and include the explanation in the caption.

Line 177. “Figure 2” must say Figure 6. Please correct.

Line 179. “There was no significant difference”. It is not possible to observe in figure 3 because it does not show the statistical treatment.

Lines 181-182. “In the second last measurement of CO2 after drying there was a weakly significant different from the rest. Otherwise,”. How is this observation achieved? It was analyzed as a factorial. Please explain.

193. Discussion.

Please discuss how the soil type (mechanism) may be affecting the performance of the sorbents.

Author Response

I write in reference to the manuscript “Does loading ammonium to sorbents affect plant availability in  soil?  by Foered and Szocs. The manuscript refers to an experiment using three different sorbents to load ammonium and fertilize a specific soil where wheat was grown. The results show that direct application of ammonium or use of  sorbents produce basically the same result. It could be interesting because sorbents may be a cheap eco-friendly form to recuperate nitrogen from alternative sources.

Although the manuscript has several problems that need to be corrected .

Specific concerns.

Lines 63 and 70. Please provide the manufacturer, address and catalog number for both products.

Bentonite is a natural product, not bought. Zeolite is bought, it does not have catalogue number, but company has been added.

Line 66 and 74. Please provide the coordinates of both sites.

Coordinates have been included.

Lines 82-83. Please describe the method or provide an appropriate reference.

Line 89. Please explain which are the characteristics of "Treatment without sorption" (table 1). Had ammonium these treatments? In what form was loaded?

It was not loaded. This has now been explained in in table caption.

Line 100. Was ammonium sulfate isotopically marked? Please explain. If it is the case, please provide the source of marked ammonium. 

This information is added, thanks for pointing out that this crucial information was missing.

Lines 111-112. “Total N and C was measured on CHN analyzer (Elementar Vario EL with TCD detector). Soil” Please describe the method or provide an appropriate reference.

Reference has been provided, now moved so it more clearly shows that it is reference for this.

Lines 112-113. “Soil and plant samples were sent to UCDa-vis for analysis of isotope ratio. Unlabelled samples were included for comparison”.  Please describe with detail the methods of analysis or provide an appropriate reference. Please explain the origin of the isotope differences.

A link to the lab where information about methods can be found is added. The origien has been clarified, see above.

Line 120. capacity. “The 0N samples were excluded from this incubation” Why?

Because very low emissions would be expected, and the main goal was to compare the different fertiliser application modes.

Line 122-125 and 133-134. Were samples taken with the bottles closed or opened? It is very confusing. Please rephrase and provide an appropriate reference.

It is now specified that that the lid was closed when the sample was taken. The second is description of the measurement in the GC-MS.

Line 140. Please indicate which PosHoc probe was used.

PostHoc was not used, only when appropriate, individual treatments were compared by ordinary t-test. This is the most correct way when the question only refers to individual treatments, not the whole group.

Line 142 “the treatment without N addition was excluded from the analysis”. Why?  Please explain which treatment, (Fig 2 only?).

It is now specified that it was excluded from the statistical analysis. This was done because we wanted to test if there was any difference between fertiliser addition modes. If we included the no fertiliser treatment, this would certainly be different.

Line 147 . Please explain (in M&M section) how was determined the origin of the nitrogen.

This has been done, as indicated above.

Lines 151-153. “When uptake from fertiliser was analysed separately, there was a significant effect of fertiliser addition and 152 sorbents.” . Does It means eliminating 0N from the statistical analysis? It would be incorrect. Please explain. 

This has been simplified, because it was already explained in materials and methods that the ANOVA was done both with and without the 0N treatment. Which result should b used depends on the question. The most interesting question here is if the way fertiliser is applied matters, and then 0N needs to be excluded.

Lines 151 156. Please rephrase, it is very confusing. 

Some rewriting as well as using the acronyms for the treatments has been done. I hope it is clearer.

All the figures, “N=3”. Please use "n" (lowercase).

Done.

Figure 1. Please use letters on the bar tops to indicate statistical differences and add the explanation in the caption including the statistical analysis employed.

Figure 2. Why aren't error bars included? Were there no replicates?

There were 3 replicates here as well, but this type of graph (stacked bars) does not allow error bars. It is also a bit difficult to see how it should be done when bars are stacked like here, you have both the individual bars, and the sum of the two, which one should error bars refer to?

Line 167. “Figure 2” must say Figure 3. Please correct. Please use letters on the bar tops to indicate statistical differences, and include the explanation in the caption.

Explanations in the captions have been included in all, and figure numbers corrected. Letters are not included, because it is not interesting to compare all treatments, and in most cases there are also no significant differences, and adding a lot of letters that are almost all the same would not improve readability of the figures.

Line 177. “Figure 2” must say Figure 6. Please correct.

Thanks, this has been corrected.

Line 179. “There was no significant difference”. It is not possible to observe in figure 3 because it does not show the statistical treatment.

No, it can’t be seen on figures, that is why the result of statistics is stated.

Lines 181-182. “In the second last measurement of CO2 after drying there was a weakly significant different from the rest. Otherwise,”. How is this observation achieved? It was analyzed as a factorial. Please explain.

This has been explained in MM that each timepoint was analysed separately.

  1. Discussion.

Please discuss how the soil type (mechanism) may be affecting the performance of the sorbents.

I believe this mostly goes outside what can be concluded from the experiment. However, some more discussion has been added, as also requested by other reviewer.

Round 2

Reviewer 1 Report

The manuscript has been revised as indicated.